# Experiences and Needs of Swiss Cancer Survivors in the Domains of Health-Related Information and the Healthcare System

**DOI:** 10.3390/cancers16244177

**Published:** 2024-12-15

**Authors:** Nicolas Sperisen, Chantal Arditi, Robin Schaffar, Pierre-Yves Dietrich, Elisabetta Rapiti

**Affiliations:** 1Institute of Global Health, Faculty of Medicine, University of Geneva, 1202 Geneva, Switzerland; 2Swiss Cancer League, 3001 Bern, Switzerland; 3Department of Epidemiology and Health Systems, Center for Primary Care and Public Health (Unisanté), University of Lausanne, 1010 Lausanne, Switzerland; chantal.arditi@unisante.ch; 4Geneva Cancer Registry, Institute of Global Health, Faculty of Medicine, University of Geneva, 1211 Geneva, Switzerland; robin.schaffar@unige.ch (R.S.); elisabetta.rapiti@unige.ch (E.R.); 5Oncology Center, Clinique des Grangettes, Hirslanden, 1224 Chêne-Bougeries, Switzerland; pierre-yves.dietrich@hirslanden.ch; 6Faculty of Medicine, University of Geneva, 1211 Geneva, Switzerland

**Keywords:** unmet needs, cancer, cancer survivors, health-related information, healthcare services, patient experiences, patient survey, PREMs, needs assessment

## Abstract

Although the acute phase of cancer treatment has ended, many survivors continue to experience lingering effects that can negatively impact their quality of life for up to 25 years after diagnosis. Approximately three quarters of these individuals encounter long-term sequelae. We investigated the experience of Swiss cancer survivors regarding health-related information and healthcare system. The Swiss healthcare system remains inadequately prepared to support cancer survivors. The current landscape of supportive care is fragmented and lacks a cohesive strategy for systematic needs assessment and tailored care plans. Gaining insights into the experiences and needs of cancer survivors in Switzerland is crucial for the development of comprehensive survivorship guidelines, which are currently lacking. Such guidelines would enhance service integration and promote a holistic approach to cancer care, ensuring that all survivors receive the necessary ongoing support throughout their recovery journey.

## 1. Introduction

### 1.1. Rationale

Cancer is one of the most frequent non-communicable diseases in Europe and the second cause of death [1]. In 2022, about 3.5 million people received a diagnosis of cancer [2]. With improvement in medical therapies and early detection, more and more people survive cancer. According to the EUROCARE-6 Working Group, approximately five percent of the European population lives through a cancer diagnosis [3]. Among these individuals, long-term survivors (cancer survivors are here defined as people living from the moment of diagnosis through to the end of life, according to the definition of the National Coalition for Cancer Survivorship (NCCS) [4]), i.e., those living five years or more after their diagnosis, represent a growing and increasingly dominant segment of the cancer survivor population [3]. Similarly, in Switzerland by the end of 2023, the estimations pointed to around 450,000 cancer survivors (about 5% of the Swiss population—2023) and a five-year survival rate estimated to be 68% [5]. Despite the improvement in survival rates, it is important to remember that cancer is a complex disease which, combined with the impact of the therapies [6,7], can have consequences and a series of negative and long-term side effects. The period following the completion of acute treatment is especially pivotal, as new and unmet needs often arise [8]. According to Miller [9], this transitional survivorship is a critical period, when people must find their new normality at the same time as experiencing physical, psychological, social, or spiritual difficulties. Cancer often becomes a chronic condition for many survivors [10,11] and can affect the whole person [12,13], their health and their quality of life [11].

Around 75% of cancer survivors encounter long-term consequences [14] that can occur many years after the diagnosis [15]. A recent scoping review [13] highlighted two important needs of long-term cancer survivors, in the domains of health-related information and the healthcare system. Health-related information needs encompass receiving accurate and timely information on all types of subjects, as well as the support necessary to understand and apply the information effectively. The healthcare system needs, on the other hand, involve the necessity for accessible, continuous, and integrated supportive care, along with appropriate and compassionate behaviour from healthcare professionals. The prominence of these needs is also underscored in the study on a group of cancer patients by Okediji et al. [16], confirming that these needs are present even before starting the treatment. Therefore, it is crucial to identify and manage these needs from the moment of diagnosis. Addressing the entire cancer journey is essential for ensuring successful survivorship and encouraging healthcare professionals to consider survivorship as an integral component of cancer care [7].

If care and treatments themselves play a significant role in the appearance of these needs, the role of patients’ experiences of care should not be underestimated. They reflect the perceived quality of received health care [17] in dimensions such as communication, support, and patient preferences [18], and are key measures to evaluate patient-centredness of care. For Ahmed et al. [19], “patient experience is an important outcome of medical care and a key component of quality of care”. Depending on people’s characteristics and experiences, values and priorities vary, but there are similarities in terms of what matters to patients [20]. Having a good experience leads to multiple positive outcomes including a smaller number of complications or adverse effects [20,21]. On the contrary, negative care experiences can contribute to the emergence of unmet needs.

### 1.2. Objective

The objectives of this study are to determine the experiences of Swiss cancer survivors in two domains, namely health-related information and healthcare systems, and their factors of influence, and to evaluate whether these experiences align with the needs identified in the literature. The following research questions were addressed:What are the experiences of Swiss cancer survivors in relation to the domains of health-related information and the healthcare system?What are the associated risk factors?Are these experiences aligning with the needs identified in the literature?

## 2. Materials and Methods

### 2.1. Data Collection

This research used secondary data collected in the Swiss Cancer Patient Experiences-2 (SCAPE-2) study, a cross-sectional, multicentred survey conducted between September 2021 and February 2022 in Switzerland [22]. SCAPE-2 collected information from cancer patients treated across eight hospitals, evenly distributed between the French- and German-speaking regions. Eligible participants were Swiss residents aged 18 and above, diagnosed with cancer, who had at least one hospitalisation or outpatient visit related to cancer between January and June 2021 at one of the participating hospitals. Data were collected with the SCAPE-2 questionnaire (Available from here https://www.scape-enquete.ch/f/questionnaire (accessed on 12 December 2024)) which consisted of 130 items, covering patient experiences with care, including experiences with information and support during and after treatment, as well as health-related and socio-demographic information. Participants had the option to complete the questionnaire on paper and return it by mail or complete it online. Of the 3220 patients who responded, 1870 indicated they had finished their treatment and were included in this research as cancer survivors.

### 2.2. Variables

Twenty-one questions addressed care experiences related to the health-related information and health system domains. Most of these questions used a 5-point scale (“yes, completely”; “yes, to some extent”; “no”; “not applicable”; and “don’t know/can’t remember”) for assessing patient experience. We computed a binary variable for each question to capture patients’ positive experiences with care (response “yes, absolutely”) versus non-positive experiences (responses “no” and “yes, to some extent”), while excluding other responses (“not applicable” and “don’t know/can’t remember”). When the proportion of individuals reporting positive experiences was 80% or less, the experience was classified as problematic. This threshold was established to reflect quality standards aligned with the high rates of positive experiences reported. It also underscores that at least one in five individuals (20%) has encountered a negative experience, highlighting unmet needs. This proportion, far from negligible, may point to systemic or structural challenges within the healthcare system that require further investigation and targeted solutions. This threshold is also used in the various analyses of the SCAPE survey [23,24]. For each question, we assigned 1 point if the patient’s experience was positive, and 0 points when it was non-positive. The overall score is computed by summing these points and dividing by the number of applicable questions, generating a percentage that reflects the quality of experiences across the selected questions for each domain. Two scores were then calculated for each domain: one score considering all the questions of the domain and one score including only problematic questions (positive experiences lower or equal to 80%). We selected 14 questions about respondents’ socio-demographic characteristics as explanatory variables to identify risk factors: geographical location, age, sex, nationality, marital status, living situation, forgoing care, difficulty paying bills, education, professional situation, follow-up period, self-rated health, cancer type, and health literacy.

### 2.3. Statistical Analysis

We conducted both univariate and multivariate regression models to assess the associations between the explanatory variables and the outcomes of interest. First, univariate linear regressions were performed for each score (continuous outcome variable), using explanatory variables to identify significant predictors. Only those variables that were significant at a *p*-value threshold of 0.1 were retained for subsequent multivariate analysis. All analyses were performed using Stata software version 18 (StataCorp, College Station, TX, USA).

## 3. Results

Sociodemographic and health characteristics of the 1870 cancer survivors included in the study are presented in Table 1. In the sample, there were slightly more women (53.9%) and people from the French-speaking region (52.7%). The majority of individuals are between 1 and 5 years post-diagnosis (39.9%), and a substantial proportion exceed 5 years (23%). About a third of the respondents considered their health to be very good or excellent (34.6%), and about half reported some difficulties in understanding written information about medical treatment or their state of health (51.7%).

Table 2 presents the responses to the 21 questions related to health-related information and the healthcare system. The experiences of Swiss cancer survivors were globally positive. For health-related information, both oral and written formats (such as information sheets, brochures, etc.) were provided to individuals. In this domain, obtaining understandable answers during consultations or outpatient treatment (91.2%) was the best rated experience, while obtaining information on late side effects (55.0%) was the worst. For the domain of the healthcare system, contact with a reference person (97.2%) was the best rated, while obtaining support from a reference person to identify solutions to their needs (54.0%) was the worst. The percentage of positive experiences was less than or equal to 80% for eleven of the twenty-one questions, six for the health-related information domain and five for the healthcare system domain, respectively. The mean scores for the pool of answers in health-related information and the healthcare system domains were 7.5 (SD 2.6) and 8.0 (SD 2.4), respectively, while the mean scores when including only the problematic experiences were 6.8 (SD 3.2) and 7.0 (SD 3.6), respectively.

Problematic experiences related to health-related information were influenced by four explanatory factors: geographical location, foregoing care, self-rated health, and health literacy; while those related to the healthcare system were influenced by three factors: foregoing care, self-rated health, and health literacy. These results are shown in Table 3. In both domains, a similar relationship was observed. Individuals who forego health care because of financial difficulties are at a higher risk for poorer care experiences. Additionally, the risk tends to increase for those with lower self-rated health and lower levels of health literacy. In addition, individuals living in French-speaking regions are at a higher risk of not receiving adequate health-related information.

## 4. Discussion

The purpose of this study was to better understand the experiences of Swiss cancer survivors in the domain of health-related information and the healthcare system. We used data from the SCAPE study that asked cancer survivors to evaluate their experiences with health care along their cancer journey. With an overall rating score of 8.5 out of ten and a range of positive experiences from 36% to 96%, cancer survivors in Switzerland reported generally positive experiences with their care [25]. This is similar to what is reported by cancer survivors in Canada [17,26,27]. However, the results also showed that, for populations in both Switzerland and Canada, there is still potential for care improvement throughout the cancer journey, including the follow-up phase. Six specific experiences in the domain of health information were deemed problematic, while five were problematic in the domain of the healthcare system.

The lower-rated aspects of health-related information highlighted unmet needs frequently identified in the literature. These include the need for better educational support, receiving adequate and clear information (that is understandable and timely), the provision of written materials in addition to verbal communication, and receiving information about long-term effects and supportive care [13]. These gaps point to the importance of improving both the delivery of information and the support provided to help patients better understand their health, in particular for people with lower health literacy who are more likely to experience problematic experiences.

Health literacy covers knowledge, motivation, and competencies to access and proceed health-related information [28]. People with a low level of health literacy also present a deteriorated state of health [29,30]. Among the SCAPE participants, half (52%) had difficulties understanding the information they had received. Helping people to improve their level of health literacy not only has an impact on their state of health but enables them to become partners in their treatment and follow-up care, and to make their own informed decisions about health care [28]. To achieve this, healthcare professionals must first provide key information at the right time [31]. Providing such information is key to encourage self-management in people with chronic illnesses [32]. Cancer patients and survivors need to receive relevant information to help them meet the challenges they face [33]. Cancer survivors therefore need to be supported by health professionals who must adapt the information to their capacity and check their understanding [31]. This underlines the vital importance of the communication skills of healthcare providers to support and empower people to take an active role in the care decision. To this end, establishing effective communication between patient and professional is crucial [34]. According to Coronado et al. [27], “negative verbal or nonverbal behaviour [on both sides] can discourage patients from participating in discussions about their care”. Martins et al. [35] agree, noting that a good relationship with a GP can make it easier for patients to seek help. The experiences of cancer survivors in our population showed that patient involvement in treatment decisions, including provision of advice and support to deal with long-term side effects, and referral to a reference person who supports them in modifying their lifestyle habits and findings solutions to their needs, are insufficient. These needs are also highlighted in the literature [13] within the domain of the healthcare system and are closely tied to the role of healthcare professionals in providing adequate involvement, support, communication, shared decision-making, and empowerment. Continuous and integrated supportive services did not emerge as significant issue in our analysis, contrasting with other Swiss studies that have emphasised gaps in support coordination [36] and a lack of a multiprofessional approach [37].

Alessy and his colleagues [17] identified in their review three categories of factors influencing experiences of care: patient’s cancer type and demographic characteristics, patient’s interactions with the healthcare system, and survey administration. Our analysis revealed four health and socio-demographic characteristics (geographical location, foregoing care, self-assessed health, and health literacy) as key factors influencing participants’ experiences. The follow-up period, or the time elapsed since diagnosis, was not found to be a significant influencing factor. Cancer survivors who stated other levels of self-rated health than excellent lived fewer positive experiences. An explanation can be that people who feel that their state of health is not optimal or who have health problems that interfere with their lives may have other needs, resulting in less satisfaction with their care [38]. Foregoing care is linked to negative experiences, suggesting economic hardship and a heightened risk of vulnerability. This aspect is underlined in our study, nearly half (43.6%) of respondents required more information on how to get financial help. Financial hardship is especially associated with lower income [39] and a lower levels of health literacy [40]. Both factors are strongly associate with low socioeconomic position (SEP) [30,41], which refers to “the social and economic factors that influence what positions individuals or groups hold within the structure of a society” [42]. SEP is a key concept to apprehend health inequalities [42,43]. The influence of SEP is well documented, and a lower SEP can negatively affect the quality of life of the person [44], as well as the quality [45] and experiences [17,46] of follow-up care. Geographical location, that is living in the French- or German-speaking regions of the country, is a powerful factor of influence, even in a small country such as Switzerland, where distances are reduced. While socio-economic status and the type of insurance has been shown not to have an influence on the treatment of breast cancer patients in Switzerland, geographical location does [47]. According to the data of the International Health Policy (IHP) Survey 2023 [48], the proportion of the response categories “excellent” or “very good” for general medical care is higher in German-speaking regions (64.4%) than in French-speaking regions of Switzerland (60.9%). Regional disparities were also identified in SCAPE data for the domain of health-related information. People who live in the German-speaking regions report more positive experiences than people who live in the French-speaking regions. Inequalities in quality of care, accessibility of care, or education of the provider could be an explanation. For Herrmann et al. [49], who reported differences in mastectomy rates between cantons or linguistic regions, variations in “health expenditures, control programmes, and treatment procedures” can be explained by the cantonal level of health policy. Cultural differences in people’s perceptions may also explain these results. The French-speaking Swiss could be more critical than the German-speaking Swiss, or providers in the German-speaking region may have better communication skills. This last point is in line with the findings of the French VICAN5 study, which identified territorial inequalities in access to information characterised by doctors’ failure to respond to specific questions [29].

Our analyses confirm that, in Switzerland, as elsewhere in the world, cancer survivors’ needs, especially regarding health-related information, are not met. It is therefore essential to regularly measure the level of these needs to be able to propose an individualised and holistic follow-up plan to respond effectively [50,51]. It is also important for healthcare professionals to be aware of the factors influencing these needs, particularly those related to socio-economic position. People from low socio-economic backgrounds may find it more difficult to access and understand health information, which can limit their involvement in care and exacerbate their needs. Consequently, a tailored and individualised approach is needed to respond appropriately to these specific challenges. Geographical location is a significant influencing factor, but its mechanisms are still poorly understood and should be explored in future research. Regional differences can affect access to care, the availability of resources, provision of information, and the quality of support offered to cancer survivors. It would be particularly relevant to include Switzerland’s third linguistic region in these studies to better understand how these factors influence patients’ needs.

This study has some limitations. First, some questions had missing data. After careful consideration of the potential use of multiple imputation techniques, we decided against imputing the missing data. Using multiple imputation to address our missing data would require modelling patients’ behaviour to synthesise unavailable data (i.e., predicting “what the patients would have answered”), relying on several assumptions for which we lack empirical basis. By trading missing values for these assumptions, we risk correcting a (possible) initial bias with a (possible) subsequent one, potentially leading to biased results. That is, multiple imputation may not provide additional information but could introduce more uncertainty into our models [52]. While multiple imputation can be a valuable approach for handling missing data, we considered the potential drawbacks of this method in our study. Unfortunately, our dataset lacks sufficient information to conduct this procedure, and we lack auxiliary variables to adequately capture the missing mechanisms. In such cases, imputation methods would rely on strong assumptions about the missing mechanisms and may produce unstable estimates. Second, the response rate of the SCAPE study was 48%, which may reflect a risk of selection bias in the study sample. Non-respondents may have been either particularly satisfied with their care or, conversely, dissatisfied. Although the risk is low, it may still negatively impact the representativeness of the sample and limit the generalisability of our findings. However, this response rate is in line with typical patient survey and is a common bias associated with these types of studies. Finally, the absence of additional socio-cultural data in the questionnaire limits the possibility of conducting a more nuanced and comprehensive analysis. This absence prevents a deeper understanding of cancer survivors’ experiences, as it neglects the influence of some socio-cultural factors that may shape how they perceive and report their care journey.

To deepen the understanding of the factors influencing cancer survivors’ experiences, future research could focus on exploring cultural variables that were not addressed in this study. Examining aspects, such as ethnic origin, religious beliefs, cultural norms and values, as well as health-related beliefs and practices, could provide valuable insights into survivors’ perceptions and experiences. A mixed-methods approach, combining both quantitative and qualitative data, alongside targeted sampling of diverse populations, would yield more comprehensive and nuanced results. Additionally, longitudinal studies could track the evolution of these factors over time and observe their impact on survivors’ perceptions of care and their healthcare journey.

## 5. Conclusions

This paper provides insights in the experiences of Swiss cancer survivors in the areas of health-related information and the healthcare system. Overall, the experiences are positive; however, lower-rated experiences highlight unmet needs. Key issues include the need for better access to high-quality information, support for understanding and processing it, and the appropriate behaviour of healthcare professionals. Addressing these aspects is crucial for optimising care quality and enhancing survivors’ quality of life. Additionally, it is important to consider the factors that influence these experiences, such as geographical location, foregoing care, self-assessed health, and health literacy. Routine assessments of survivors’ needs and experiences could help achieve these goals. Addressing these challenges through policy measures in Switzerland could significantly enhance the outcomes of cancer treatments.

## Figures and Tables

**Table 1 cancers-16-04177-t001:** Sociodemographic and health characteristics of the respondents.

Variables	Frequence *n* (%)
**Sex (*n =* 1846)** Women Men	995 (53.9%) 851 (46.1%)
**Geographical Location (*n* = 1870)** German-speaking region French-speaking region	884 (47.3%) 986 (52.7%)
**Follow-up period (*n =* 1870)** The year after treatment ends 1 to 5 years after the end of treatment 5 years or more after the end of treatment	694 (37.1%) 746 (39.9%) 430 (23%)
**Age (*n =* 1755)** 18–39 40–64 65–79 80+	117 (6.7%) 756 (43.1%) 740 (42.2%) 142 (8.1%)
**Education (*n =* 1802)** No schooling/compulsory school Apprenticeship (CFC), vocational school, basic vocational training General education school, Gymnasium/prof. matriculation, Teacher training college/pedagogic school Advanced technical and professional training/higher technical or commercial college University, university of applied sciences, federal institute of technology	203 (11.3%) 574 (31.9%) 202 (11.2%) 458 (25.4%) 365 (20.3%)
**Foregoing care (*n =* 1819)** Yes	144 (7.7%)
**Difficulty paying bills (*n =* 1827)** Yes	205 (11.22%)
**Self-rated health (*n =* 1819)** Excellent Very good Good Mediocre Bad	129 (7.1%) 500 (27.5%) 970 (53.3%) 202 (11.1%) 18 (1%)
**Health literacy (*n =* 1818)** Never Occasionally Sometimes Often Always	878 (48.3%) 504 (27.7%) 336 (18.5%) 70 (3.9%) 30 (1.7%)

**Table 2 cancers-16-04177-t002:** Scores, percentage, and number of positive experiences relating to health-related information and the healthcare system domains.

Domain	Questions	% (*n*) of Positive Experience
Health-related information	1. Received clear and understandable answers to questions	91.2 (1277)
	2. Received information on cancer’s impact on daily activities	84.0 (1081)
	3. Received information on support groups	82.3 (731)
	4. Received written information for post-discharge period	82.0 (433)
	5. Received needed information on follow-up	80.1 (1458)
	6. Received information on support options to manage emotions	78.7 (709)
	7. Understood diagnostic explanations	74.5 (1362)
	8. Treatment side effects explained in understandable way	73.6 (1312)
	9. Received information on how to get financial help	56.4 (397)
	10. Received written diagnostic information	56.4 (674)
	11. Informed on long-term side effects	55.0 (918)
	Mean score for questions 1 to 11	7.5 (SD 2.6)
	Mean score for problematic questions (questions 6 to 11)	6.8 (3.2)
Healthcare system	12. In regular contact with a reference person for follow-up	97.2 (1504)
	13. GP received sufficient information on health and treatment	92.3 (1454)
	14. Professionals worked well together for optimal treatment	85.4 (1539)
	15. Received a specialized nursing consultation	82.9 (582)
	16. Received support from a reference person to deal with symptoms	80.8 (1066)
	17. Received a care plan	80.0 (712)
	18. Involved in treatment decisions as desired	74.4 (1345)
	19. Offered advice/support to deal with long-term effects	64.2 (728)
	20. Received support from a reference person to modify lifestyle habits	56.5 (459)
	21. Received support from a reference person to identify solutions to their needs	54.0 (378)
	Mean score for questions 12 to 21	8.0 (SD 2.4)
	Mean score for problematic questions (question 17 to 21)	7.0 (SD 3.6)

**Table 3 cancers-16-04177-t003:** Explanatory factors significantly associated with problematic experience scores in the areas of health-related information and the healthcare system, in multivariate linear regressions.

Explanatory Factors	Health-Related Information *			Healthcare System ^†^		
	Beta	95% CI	*p*-Value	Beta	95% CI	*p*-Value
Geographical location			0.026			
German-speaking region	Ref					
French-speaking region	−3.4	−6.4, −0.42	0.026			
Foregoing care			<0.001			<0.001
No	Ref			Ref		
Yes	−11	−17, −4.6	<0.001	−12	−19, −4.9	<0.001
Self-rated health			<0.001			<0.001
Excellent	Ref			Ref		
Very good	−3	−9.0, 3.1	0.3	−10	−17, −2.6	0.008
Good	−9.9	−16, −4.0	0.001	−16	−23, −9.0	<0.001
Mediocre	−19	−26, −11	<0.001	−29	−38, −20	<0.001
Bad	−17	−33, −0.67	0.041	−40	−60, −20	<0.001
Health literacy			<0.001			<0.001
Never	Ref			Ref		
Occasionally	−11	−15, −8.0	<0.001	−5.7	−9.9, −1.5	0.008
Sometimes	−14	−18, −9.9	<0.001	−10	−15, −5.3	<0.001
Often	−25	−32, −17	<0.001	−17	−26, −7.6	<0.001
Always	−10	−23, 2.6	0.12	1.2	−14, 17	0.9

Statistically significant values are indicated in bold. Ref: reference category. * Adjusted for geographical location, foregoing care, self-rated health and health literacy. **^†^** Adjusted for foregoing care, self-rated health and health literacy.

## Data Availability

While the dataset generated and analysed in this study is not publicly available, it can be accessed upon reasonable request from the data.unisante.ch repository (DOI: https://doi.org/10.16909/DATASET/38).

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
