# Peer review of "Experiences and Needs of Swiss Cancer Survivors in the Domains of Health-Related Information and the Healthcare System"

_cancers, 2024, doi:10.3390/cancers16244177_

Round 1
Reviewer 1 Report
Comments and Suggestions for Authors
Overall this is a good paper, well-written and relevant. The response rate of 48% is actually pretty good, and I doubt whether a larger survey could have got better results. The risk of bias is quite low as survey reports go. In terms of content, there is no mention of how long the survivors were out from their diagnosis or ending of treatment. If the authors have this information, it would be helpful, as needs may vary depending on the stage of survivorship. Also, the final sentence of the conclusion is perhaps a bit open to debate; “Addressing these concerns at the policy level in Switzerland is essential.” Routine assessment of cancer survivors would be expensive and require system-wide responses, which though clearly desirable, may not be a priority when looked at in the context of limited resources and many other needs: like universal access to high quality palliative care, shorter wait times for assessment, imaging and treatment, and access to certain expensive drugs. I would perhaps re-word this last sentence as “ Addressing these concerns at a policy level in Switzerland would contribute to better outcomes of cancer treatment” or something similar.
There are some translation issues or typos which are very minor but would be good to have corrected if possible. I have listed those I have spotted below.
Line 57, there is an apostrophe instead of a comma on 450’000 cancer survivors (it’s normally 450,000)
In line 201-2 the sentence “Among the SCAPE participants, one cancer survivor 201 out of two (52%) has difficulties to understand the information received” is a little clumsy, suggest re-word as “Among the SCAPE participants, half (52%) had difficulties understanding the information they had received.” Similarly in lines 237-8 “….with almost one person out of two (43.6%) requiring more information on how to get financial help” might read better as “nearly half (43.6%) of respondents required more information on how to get financial help.”
Line 239 “Financial hardship is especially associated to lower income” should be “…with lower income”.
Line 240 “Both factors are strongly associate with a low socioeconomic position” should be “….associated with low…..”.
Line 261 “…may have better communication’s skills.” Should be “…….communication skills.”
Thankyou for the opportunity to review this paper.
Author Response
Dear Reviewer 1,
Thank you for taking the time to review our article titled " Experiences and needs of Swiss cancer survivors in the domains of health-related information and healthcare system". We appreciate your insightful feedback, which has greatly contributed to the enhancement of our work.
Attached, you will find the revised version of the manuscript, where we have addressed each of your comments and suggestions in detail. Below, we provide a summary of the changes made:
- We have completed Table 1 (refer to line 137) and the text (refer to lines 132-133, 209-210) with the information available on the follow-up period.
- We have changed the last sentence of the conclusion as you suggested (refer to lines 280-281).
- We have undertaken the corrections you have suggested (refer to lines 54, 186, 214-216, 232-233).
We believe that these revisions have significantly improved the clarity, coherence, and overall quality of the manuscript. We are grateful for the opportunity to address your concerns and refine our work accordingly.
Should you have any further questions or require additional clarification, please do not hesitate to contact us.
Thank you once again for your valuable feedback and constructive suggestions.
Best regards,
Nicolas Sperisen
Reviewer 2 Report
Comments and Suggestions for Authors
Interesting analysis regarding adult patient experiences concerning on care, information and support during and after cancer treatment. In general, the results indicate that cancer survivors in Switzerland received satisfactory information and help. In my opinion , it would be interesting to indicate what were the sources of information about the disease and late effects (oral information, booklets ). Does age at the time of therapy, the time that has passed since the end of treatment and current age have an impact on the assessment of health status? Aging and comorbidities can significantly impact self-assessment of health, needs and quality of life. Older people > 65 years of age constituted 50% of the respondents, often they do not understand the questions being asked, (and they do not want to admit it), they have other problems, including financial ones.
It would be useful to include a link to the questionnaire.
Author Response
Dear Reviewer 2,
Thank you for taking the time to review our article titled " Experiences and needs of Swiss cancer survivors in the domains of health-related information and healthcare system". We appreciate your insightful feedback, which has greatly contributed to the enhancement of our work.
Attached, you will find the revised version of the manuscript, where we have addressed each of your comments and suggestions in detail. Below, we provide a summary of the changes made:
- Unfortunately, the format of the information sources could not be determined. The evaluation focused on participants’ experiences with receiving information (such as whether they received it (yes/no), whether it was understandable, or whether written information was provided specifically for the diagnosis) rather than the type of source itself. However, in general, these sources tend to be varied. We have clarified this point and added details in the text (refer to lines 140-141).
- We acknowledge that aging and comorbidities can affect the self-assessment of health and quality of life in older adults. However, we used the 1st item of the SF-36 and the Fact-g7 scale, which have both been validated in older patients and shown strong reliability and validity in this population (ref: Overcash J et al. Validity and reliability of the FACT-G scale for use in the older person with cancer. Am J Clin Oncol. 2001;24(6):591-6.; Walters SJ et al. Using the SF-36 with older adults: a cross-sectional community-based survey. Age Ageing. 2001;30(4):337-43).
Psychological studies have demonstrated that individuals' responses to subjective questions about health or quality of life tend to shift with age, influenced by various mechanisms:
- According to the Selection, Optimization, and Compensation model (ref: Baltes, P. B., & Baltes, M. M. (1990). Psychological perspectives on successful aging: The model of selective optimization with compensation. In P. B. Baltes & M. M. Baltes (Eds.), Successful aging: Perspectives from the behavioral sciences(pp. 1–34). Cambridge University Press. https://doi.org/10.1017/CBO9780511665684.003; Baltes, M. M., & Lang, F. R. (1997). Everyday functioning and successful aging: The impact of resources. Psychology and Aging, 12(3), 433–443. https://doi.org/10.1037/0882-7974.12.3.433), individuals adapt by reevaluating their priorities to address challenges and optimize available resources. Consequently, even if some areas of their lives worsen objectively, they may channel their focus into more meaningful domains, allowing them to perceive their quality of life as stable or even improved.
- Furthermore, the theory of scale recalibration (ref: Blome, C., & Augustin, M. (2015). Measuring change in quality of life: Bias in prospective and retrospective evaluation. Value in Health, 18(1), 110–115. https://doi.org/10.1016/j.jval.2014.10.007) suggests that people's internal benchmarks for interpreting response scales evolve over time. For instance, between ages 60 and 80, greater exposure to individuals with significant health issues may unconsciously shift these benchmarks, resulting in a reassessment of their own health or quality of life that reflects this adjusted perspective.
Regarding co-morbidities, it is indeed true that they impact the measurement of health. Typically, single-item health questions are posed without defining the concept of health explicitly. This approach allows respondents to interpret the question according to their personal priorities and values, encompassing the aspects of health they consider most relevant. There is indeed a strong association between perceived health and the presence of co-morbidities, as co-morbidities are inherently linked to an individual's overall health status. However, we do not see this as a potential bias. Instead, it represents a fundamental aspect of assessing perceived health holistically, reflecting the true complexity of an individual’s health experience.
- We have included a link to the questionnaire (refer to line 99 – footnote).
We believe that these revisions have significantly improved the clarity, coherence, and overall quality of the manuscript. We are grateful for the opportunity to address your concerns and refine our work accordingly.
Should you have any further questions or require additional clarification, please do not hesitate to contact us.
Thank you once again for your valuable feedback and constructive suggestions.
Best regards,
Nicolas Sperisen
Reviewer 3 Report
Comments and Suggestions for Authors
This manuscript presents a valuable contribution to the understanding of cancer survivors in Switzerland, offering insights with significant potential for advancement in health research. We propose the following recommendations to enhance the scientific rigor and impact of the study:
1) The current study employs an 80% or lower threshold; however, the statistical and scientific rationale underlying this specific threshold remains insufficiently elucidated. We recommend a comprehensive exposition of the methodological considerations that informed this threshold selection, including potential evidence-based justifications and statistical validation.
2) The current treatment of missing data, specifically the decision to eschew multiple imputation technique, requires more explicit substantiation within the methodological section. A detailed explication of the statistical reasoning, potential implications for data integrity, and the potential impact on research outcomes is strongly recommended.
3)While the current analysis acknowledges potential sampling biases, a more nuanced exploration incorporating cultural dimensions would substantially augment the analytical depth. Integrating sociocultural contextual factors could provide a more comprehensive understanding of the observed variations and potential limitations.
4)The concluding section would be significantly strengthened by a more expansive discussion of prospective research directions. A forward-looking perspective that articulates potential intervention strategies, research methodologies, and anticipated knowledge gaps would enhance the manuscript's scholarly contribution and future research potential.
Author Response
Dear Reviewer 3,
Thank you for taking the time to review our article titled " Experiences and needs of Swiss cancer survivors in the domains of health-related information and healthcare system". We appreciate your insightful feedback, which has greatly contributed to the enhancement of our work.
Attached, you will find the revised version of the manuscript, where we have addressed each of your comments and suggestions in detail. Below, we provide a summary of the changes made:
- As recommended, we have supplemented the methodology section with a more detailed explanation of the reasons for choosing this threshold (refer to lines 110-115).
- As recommended, we have completed the explanation of our choice for handling missing data (refer to lines 248-258).
- Due to the lack of additional cultural dimensions in the questionnaire, we are unable to produce more detailed data on this subject. We have completed the limitations section with this point (refer to lines 262-265).
- As recommended, we have suggested a direction for future prospective research focused on the cultural dimensions highlighted in the section above (refer to lines 267-272).
We believe that these revisions have significantly improved the clarity, coherence, and overall quality of the manuscript. We are grateful for the opportunity to address your concerns and refine our work accordingly.
Should you have any further questions or require additional clarification, please do not hesitate to contact us.
Thank you once again for your valuable feedback and constructive suggestions.
Best regards,
Nicolas Sperisen